# Analysis of SARS-CoV-2 Screening Clinic (Including Drive-through System) Data at a Single University Hospital in South Korea from 27 January 2020 to 31 March 2020 During the COVID-19 Outbreak

**DOI:** 10.3390/healthcare8020145

**Published:** 2020-05-26

**Authors:** Min Cheol Chang, Wan-Seok Seo, Donghwi Park, Jian Hur

**Affiliations:** 1Department of Physical Medicine and Rehabilitation, College of Medicine, Yeungnam University, Daegu 42415, Korea; wheel633@ynu.ac.kr; 2Department of Psychiatry, College of Medicine, Yeungnam University, Daegu 42415, Korea; sws3901@ynu.ac.kr; 3Department of Physical Medicine and Rehabilitation, Ulsan University Hospital, University of Ulsan College of Medicine, Ulsan 44033, Korea; 4Department of Infectious Disease Internal Medicine, College of Medicine, Yeungnam University, Daegu 42415, Korea

**Keywords:** COVID-19, drive-through screening system, real-time polymerase chain reaction

## Abstract

In this study, we evaluated the efficiency of a drive-through (DT) screening system for severe acute respiratory syndrome coronavirus 2 (SARS-CoV-2) by comparing it with a conventional screening system. We reviewed and analyzed the SARS-CoV-2 screening data obtained at our university hospital. We compared the number of tests for SARS-CoV-2 (using real-time polymerase chain reaction) performed using two different specimen collection systems—DT and conventional—during the coronavirus disease 2019 (COVID-19) outbreak in Daegu. Based on the results, the DT screening system collected 5.8 times more specimens for testing than the conventional screening system. From 27 January to 31 March 2020, 6211 individuals were screened for SARS-CoV-2 infection using either the DT or conventional system. In total, 217 individuals tested positive for SARS-CoV-2 (positive rate: 3.50%). Of the 6211 individuals, 3368 were symptomatic or had a history of contact with COVID-19 patients, and 142 of them tested positive for SARS-CoV-2 (positive rate: 4.22%). Further, 2843 individuals were asymptomatic and had no history of contact with COVID-19 patients, and 75 of them tested positive for SARS-CoV-2 (positive rate: 2.64%). In conclusion, the DT system allowed clinicians to collect specimens for SARS-CoV-2 screening more efficiently than the conventional system. Furthermore, as there might be several COVID-19 patients who remain asymptomatic, expanding the screening test to asymptomatic individuals would be necessary.

## 1. Introduction

After the first reported case of coronavirus disease 2019 (COVID-19) in Hubei Province of China in December 2019, COVID-19 cases have been reported in most countries worldwide. The novel severe acute respiratory syndrome coronavirus 2 (SARS-CoV-2, previously known as 2019-nCoV) has been rapidly spreading, with unprecedented propagation, because of its highly infectious nature. The World Health Organization declared the COVID-19 outbreak a pandemic on 11 March 2020 [1,2,3]. The fatality rate of COVID-19 is 2–6% and is much higher in older populations and those with underlying diseases [1,2,3]. The number of suspected or symptomatic individuals with COVID-19 is continuously increasing; hence, screening clinics, separated from other existing clinics, are being set up in several hospitals in each community.

In Daegu, South Korea, a rapid surge in COVID-19 cases occurred in late February and early March 2020. Over 5000 confirmed cases of COVID-19 were reported during this period in Daegu. For efficient and safe screening, some hospitals in Daegu implemented a drive-through (DT) screening system [4]. The individuals who used this system could provide a sample for SARS-CoV-2 testing without leaving their cars. We believe that this system is helpful for conducting rapid and safe testing during the current COVID-19 outbreak in Daegu. 

In the current study, we evaluated the efficiency of a DT screening system for SARS-CoV-2 by comparing it with a conventional screening system. Moreover, we reviewed the results of SARS-CoV-2 screening (for samples collected via DT and conventional systems) in a single university hospital during the COVID-19 outbreak in Daegu, South Korea. 

## 2. Methods

This study was approved by the Institutional Review Board (IRB) of Yeungnam University Hospital, and the requirement for informed consent was waived by the Ethics Committee. We reviewed and analyzed the data of a SARS-CoV-2 screening clinic at Yeungnam University Hospital, Daegu, which is one of the four university hospitals in Daegu. The data were collected using a computerized system. At the time of patient examination, the staff input the data, such as the presence of COVID-19 symptoms and a history of contact with COVID-19 patients, into the hospital’s computerized system. In addition, we described the details of the DT screening system and the conventional screening system and compared the two systems. This research was approved by IRB at Yeungnam University Hospital (2020-03-101).

## 3. Results

### 3.1. Operational Differences between the Conventional and Drive-Through Systems

#### 3.1.1. Conventional System

The conventional specimen collection system used a negative-pressure tent. The medical staff working in the temporary buildings wore personal protective equipment, including inner and outer gloves; an N95 respirator; an eye shield, a face shield, or goggles; a hooded coverall or gown. The outer gloves and disposable plastic apron (AP) gowns were changed after contact with every patient. The tent was divided into three sections—the section through which the individual to be tested entered (Section A), the section where the medical staff completed the questionnaires (Section B), and the section where the specimen for testing was collected (Section C) (Figure 1). When a testee entered the negative-pressure tent (Section A), the medical staff in Section B interviewed the testee. These two sections (A and B) were separated by transparent plastic. After completing the questionnaire, the testee moved to Section C, where the test specimens were collected. In Section C, three or four medical staff were waiting—one was responsible for collecting the payment for the test, one was responsible for specimen collection, and the others were responsible for sterilizing the space after the individual exited the tent. To prevent cross-contamination by testees, all sections, except Section B, were sterilized for at least 30 minutes before the next testee entered.

#### 3.1.2. Drive-Through System

The procedure for the DT specimen collection system was as follows: entrance, registration and questionnaire completion, examination, specimen collection, and receipt of instructions and information (Figure 2) [4]. For registration and questionnaire completion, examination, specimen collection, and receipt of instructions and the information leaflet, one temporary outside building was built for each section; therefore, four temporary buildings were built in total. One or two medical staff were present in each section and were completely isolated from the outdoor environment. The medical staff working in the temporary buildings wore personal protective equipment, including inner and outer gloves; an N95 respirator; an eye shield, a face shield, or goggles; a hooded coverall or gown. The outer gloves and disposable plastic apron (AP) gowns were changed after contact with every patient. In the registration and questionnaire section, the identification card of the testee was scanned and automatically transferred to the computerized system of the hospital to avoid direct contact with testees. In the registration and questionnaire and examination sections, microphones were installed inside and outside of the temporary buildings so that the medical staff could communicate with the testee without direct contact. In the specimen collection section, the driver opened the window of the car and collected his or her own specimen. In the instructions and information section, the medical staff provided information regarding when the results would be available through a microphone. The entire service could be provided to individuals without leaving their cars. Unlike conventional systems, the DT system did not require sterilization of spaces between testees, thus markedly reducing the preparation times. The total time spent on each testee was 5–7 min. Comparing the time taken for the COVID-19 test between the DT system (5–7 min) and conventional system (30 min), the DT system was 4–6 times more efficient than the conventional screening system.

### 3.2. The Difference in Specimen Collection Ability between Drive-Through and Conventional Systems

We compared the number of specimens collected by the two different systems over a 14-day period (26 February 2020 to 10 March 2020), during which both systems were operating simultaneously. Through this, we aimed to indirectly compare the efficiency of the two systems. In the DT and conventional systems, the number of specimens collected per day were 241.7 and 41.4, respectively (3384 (DT system) vs. 580 (conventional system)). According to the total number of tests performed during the 14-day period, the efficiency of the DT system was 5.8 times higher than that of the conventional screening system. In addition, per test, the DT system was 4–6 times faster than the conventional system (5–7 min vs. 30 min). Because of the higher efficiency of the DT system, as of 11 March 2020, we stopped using the conventional system and used only the DT system to collect specimens for SARS-CoV-2 testing.

### 3.3. Laboratory Data

Real-time polymerase chain reaction (Allplex^™^ 2019-nCoV Assay, Seegene^®^, Seoul, Korea) was used for SARS-CoV-2 testing. From 17 January to mid-February 2020, this test was performed in less than 10 individuals per day. The first confirmed case of COVID-19 was reported on 18 February 2020, and a gradual increase in the number of tests and confirmed cases followed. After the sudden outbreak of COVID-19 in Daegu, the number of tests, newly confirmed cases, and positive rate sharply increased. In mid-March 2020, 150–450 individuals were being tested for SARS-CoV-2 daily (Figure 3 and Table 1).

In total, 6211 individuals underwent SARS-CoV-2 testing. Overall, 5151 (82.93%) and 1060 (17.07%) individuals provided specimens through DT and conventional collection systems, respectively. Further, 217 individuals tested positive for SARS-CoV-2 (positive rate: 3.50%). Of the 6211 individuals, 3368 were symptomatic or had a history of contact with COVID-19 patients, and 142 of them tested positive for SARS-CoV-2 (positive rate: 4.22%). Further, 2843 patients were asymptomatic and had no history of contact with COVID-19 patients, and 75 of them tested positive for SARS-CoV-2 (positive rate: 2.64%) (Table 2). A large number of individuals aged 20 to 59 years, >1000 per age range, underwent SARS-CoV-2 testing, and the positive rate was 1.5–4.6% (Table 3). In contrast, a relatively small number of individuals aged 60 to 89 years underwent SARS-CoV-2 testing, but the positive rate was relatively high (6.8–14.0%). Of the 211 patients with positive test results, 89 (41.01%) were men, and 128 (58.99%) were women. 

During the 14-day period (26 February 2020 to 10 March 2020), when DT and conventional systems were administered simultaneously, 3964 individuals underwent SARS-CoV-2 testing. According to the screening system, the rates of positive test results were 4.11% (DT system) and 5.69% (conventional system). Of the 3384 individuals who used the DT system, 1812 (53.55%) were symptomatic or had a history of contact with COVID-19 patients, while 1572 (46.45%) were asymptomatic and had no history of contact with COVID-19 patients. Of the 580 individuals who used the conventional system, 332 (57.24%) were symptomatic or had a history of contact with COVID-19 patients, while 248 (42.76%) were asymptomatic and had no history of contact with COVID-19 patients (Table 4).

## 4. Discussion

For managing pandemic diseases, such as COVID-19, not only accuracy but also the efficiency of diagnoses, such as the time taken to establish a diagnosis and the number of tests performed, are of utmost importance. Since SARS-CoV-2 is highly transmissible, rapid diagnosis and sufficient testing capacity are essential for the management of COVID-19 [1,2,3]. In our study, two specimen collection systems—conventional and DT—were examined. The DT system was approximately 9 times more efficient at collecting specimens for testing. These results suggested that the DT system was more efficient and useful for effective COVID-19 management. We believed that the wide-scale use of DT systems could significantly contribute to the efficient diagnosis of COVID-19. 

In this study, positive results were seen in around 2.64% of asymptomatic individuals. These results supported that lack of symptoms should not be an indication of exclusion for SARS-CoV-2 testing. In other words, testing might be necessary during pandemics, even when if the patient is asymptomatic or has a history of contact with infected individuals [5,6]. 

In our study, we found that older individuals (aged ≥ 60 years) had a lesser tendency to undergo screening than younger individuals (aged ≤ 50 years). The positive test result rates were higher among individuals in their 60 s–80 s than among those in the lower age groups. Older individuals had a lesser tendency to undergo the COVID-19 test than younger individuals. This could be due to the fact that many older individuals are physically weak or cannot drive. To develop appropriate strategies for managing COVID-19, the government and clinicians must focus more on the older population, especially individuals aged ≥ 60 years. 

During the study period, the number of tests and the cumulative number of positive patients began to increase exponentially from 20 February 2020. The main source of this rapid propagation was determined to be mass infection among a religious group (Shincheonji) in Daegu during a Sunday gathering. In this religious gathering, thousands of people were sitting close to each other in a confined space within the facility, having conversations, praying, and singing songs. The government had ordered screening tests for all the individuals in this group. Of them, 62% (>2000 cases) tested positive for SARS-CoV-2. Moreover, throughout South Korea, mass infections occurring in churches, hospitals, nursing homes, and call centers were largely responsible for the increase in the number of COVID-19 cases. Therefore, during the COVID-19 outbreak, group meetings should be restrained, and measures should be implemented in facilities where many people are gathered to actively prevent further COVID-19 cases as SARS-CoV-2 has a high transmission rate.

As of 11 April 2020, the total positive test result rate for SARS-CoV-2 in South Korea was 2.05% (http://ncov.mohw.go.kr/en/). This value was lower than the positive test result rate of our study (3.49%). This might be due to the fact that our hospital is located in Daegu, which was the epicenter of the COVID-19 outbreak in South Korea; as of 11 April 2020, around 65% of COVID-19 patients in South Korea were Daegu residents.

## 5. Conclusions

In this study, we reviewed screening clinic data for SARS-CoV-2 in a single university hospital during the COVID-19 outbreak in Daegu, South Korea. The results of the analysis showed that the DT screening system was highly efficient. The DT system was a useful option for rapid and safe testing. Furthermore, our results showed a high positive test result rate in asymptomatic patients. Therefore, expanding screening tests to asymptomatic individuals might be necessary. Larger cohort studies analyzing data from multiple hospitals are warranted for acquiring more accurate information regarding SARS-CoV-2 and COVID-19. 

## Figures and Tables

**Figure 1 healthcare-08-00145-f001:**
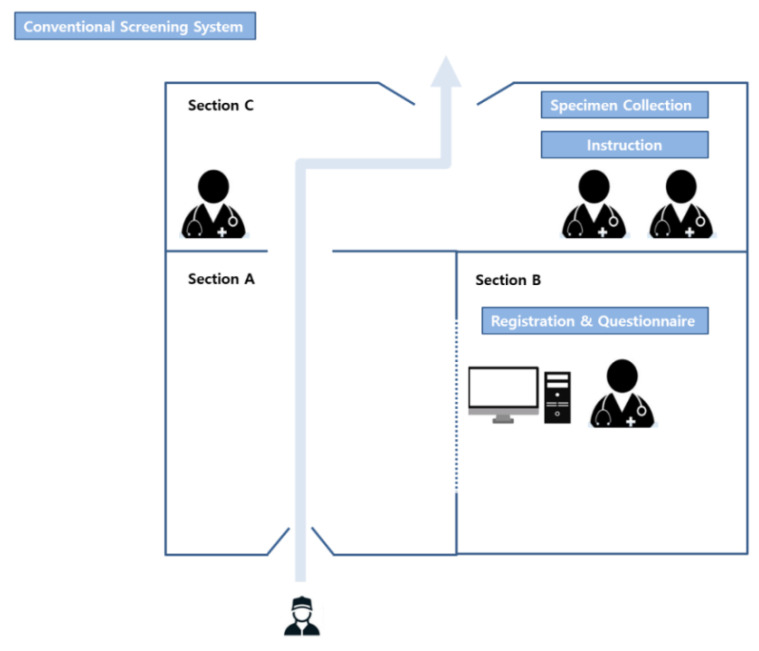
Illustration of the conventional severe acute respiratory syndrome coronavirus 2 (SARS-CoV-2) screening process.

**Figure 2 healthcare-08-00145-f002:**
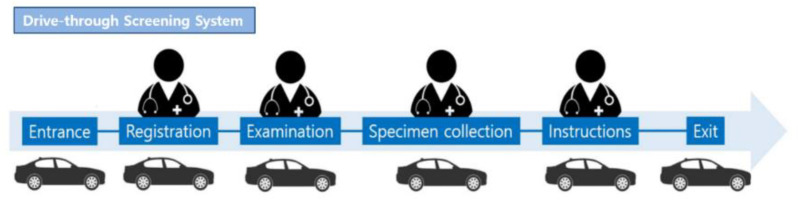
Illustration of the drive-through SARS-CoV-2 screening process.

**Figure 3 healthcare-08-00145-f003:**
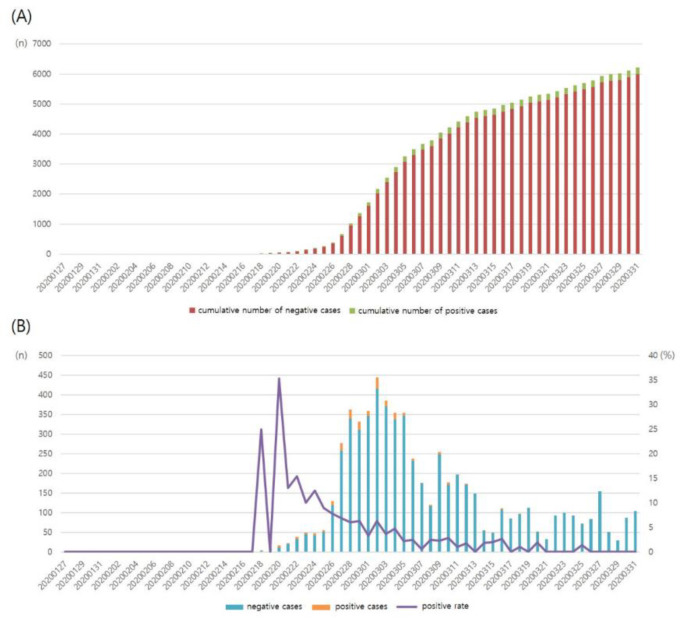
(**A**) Graph showing the cumulative number of positive and negative cases after SARS-CoV-2 testing and positive test result rate by date in the screening clinic of our university hospital. (**B**) Graph showing the daily number of positive and negative cases after SARS-CoV-2 testing.

**Table 1 healthcare-08-00145-t001:** A cumulative number of positive and negative severe acute respiratory syndrome coronavirus 2 (SARS-CoV-2) cases and the number of positive and negative cases by date in the screening clinic of our university hospital.

Date (YYYYMMDD)	Number of Tested Cases	Cumulative Number of Negative Cases	Cumulative Number of Positive Cases	Positive Rate	Number of Negative Cases Per Day	Number of Positive Cases Per Day
20200127	0	0	0	-	0	0
20200128	0	0	0	-	0	0
20200129	1	1	0	0	1	0
20200130	1	2	0	0	1	0
20200131	0	2	0	-	0	0
20200201	1	3	0	0	1	0
20200202	0	3	0	-	0	0
20200203	0	3	0	-	0	0
20200204	0	3	0	-	0	0
20200205	0	3	0	-	0	0
20200206	0	3	0	-	0	0
20200207	0	3	0	-	0	0
20200208	1	4	0	0	1	0
20200209	0	4	0	-	0	0
20200210	3	7	0	0	3	0
20200211	2	9	0	0	2	0
20200212	1	10	0	0	1	0
20200213	0	10	0	-	0	0
20200214	1	11	0	0	1	0
20200215	0	11	0	-	0	0
20200216	0	11	0	-	0	0
20200217	1	12	0	0	1	0
20200218	4	15	1	25.00	3	1
20200219	11	26	1	0	11	0
20200220	17	37	7	35.29	11	6
20200221	23	57	10	13.04	20	3
20200222	39	90	16	15.38	33	6
20200223	50	135	21	10.00	45	5
20200224	48	177	27	12.50	42	6
20200225	56	228	32	8.93	51	5
20200226	129	347	42	7.75	119	10
20200227	277	605	61	6.86	258	19
20200228	362	945	83	6.08	340	22
20200229	332	1256	104	6.33	311	21
20200301	359	1603	116	3.34	347	12
20200302	444	2019	144	6.31	416	28
20200303	385	2390	158	3.64	371	14
20200304	355	2728	175	4.79	338	17
20200305	355	3075	183	2.25	347	8
20200306	238	3307	189	2.52	232	6
20200307	176	3482	190	0.57	175	1
20200308	120	3599	193	2.50	117	3
20200309	255	3848	199	2.35	249	6
20200310	177	4020	204	2.82	172	5
20200311	198	4216	206	1.01	196	2
20200312	174	4387	209	1.72	171	3
20200313	149	4536	209	0	149	0
20200314	55	4590	210	1.82	54	1
20200315	50	4639	211	2.00	49	1
20200316	111	4747	214	2.70	108	3
20200317	85	4832	214	0	85	0
20200318	98	4929	215	1.02	97	1
20200319	113	5042	215	0	113	0
20200320	52	5093	216	1.92	51	1
20200321	33	5126	216	0	33	0
20200322	93	5219	216	0	93	0
20200323	100	5319	216	0	100	0
20200324	93	5412	216	0	93	0
20200325	73	5484	217	1.37	72	1
20200326	84	5568	217	0	84	0
20200327	155	5723	217	0	155	0
20200328	51	5774	217	0	51	0
20200329	29	5803	217	0	29	0
20200330	87	5890	217	0	87	0
20200331	104	5994	217	0	104	0

**Table 2 healthcare-08-00145-t002:** The number of positive SARS-CoV-2 cases and positive rate according to the presence of symptoms and history of contact with coronavirus disease 2019 (COVID-19) patients.

Classification of COVID-19 Patients According to Symptom or History of Contact	Number of Tested Individuals (*n*)	Number of Positive Cases (*n*)	Positive Rate (%)
Symptomatic or having a history of contact with COVID-19 patients	3368	142	4.22
Asymptomatic and having no history of contact with COVID-19 patients	2843	75	2.64

**Table 3 healthcare-08-00145-t003:** The number of tested individuals, number of SARS-CoV-2-negative and -positive cases, and positive rates according to age ranges.

Age (Years)	Number of Tested Individuals (*n*)	Number of Negative Cases (*n*)	Number of Positive Cases (*n*)	Positive Rate (%)
<10	103	101	2	1.94
10s	142	139	3	2.11
20s	1217	1185	32	2.63
30s	1333	1313	20	1.50
40s	1299	1271	28	2.16
50s	1219	1164	55	4.51
60s	628	585	43	6.85
70s	193	166	27	13.99
80s	68	61	7	10.29
90s	9	9	0	0

**Table 4 healthcare-08-00145-t004:** The number of SARS-CoV-2-positive cases and positive rates based on the presence of symptoms and history of contact with COVID-19 patients according to the screening system.

Classification of COVID-19 Patients According to Symptom or History of Contact	Drive-Through System	Conventional System	Total
Symptomatic or having a history of contact with COVID-19 patients	1812 (53.55%)	332 (57.24%)	2144 (54.09%)
Asymptomatic and having no history of contact with COVID-19 patients	1572 (46.45%)	248 (42.76%)	1820 (45.91%)
Positive for COVID-19	139 (4.11%)	33 (5.69%)	172 (4.34%)

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
