# Peer review of "Analysis of SARS-CoV-2 Screening Clinic (Including Drive-through System) Data at a Single University Hospital in South Korea from 27 January 2020 to 31 March 2020 During the COVID-19 Outbreak"

_healthcare, 2020, doi:10.3390/healthcare8020145_

Round 1

Reviewer 1 Report

This is a timely report and has sufficient sample size.

However, an analysis  of detection accuracy of DT is needed.

From Feb.26 to Mar.10, how patients were divided into DT or conventional sample collection? Was it random?

What is the positive rate for DT group and conventional group during this period. 

The evaluation of both efficiency and accuracy are needed.

1, The evaluation of efficiency includes the cost performance of each procedure in addition to the number of tests. Describe the usage of PPE (personal protective equipment: mask, face shield, gloves, and gown), and compare their per test between DT and conventional specimen collection system.

2, The accuracy of DT system may become clear by comparing the positive ratio among the following 4 groups in the 14-day period (from February 26, 2020, to March 10, 2020) during which both systems were operating simultaneously.

  • Symptomatic-DT
  • Symptomatic-conventional
  • Asymptomatic-DT
  • Asymptomatic-conventional

The DT procedure is accurate when 1)=2), and 3)=4).

Author Response

Reviewer1

This is a timely report and has sufficient sample size.

Answer: We appreciate your valuable comments.

However, an analysis of detection accuracy of DT is needed.

Answer: We appreciate your valuable comment. We totally agree with your comment. Following your comment, we have added in detection accuracy of both DT and conventional system in Table 4.

However, in both of DT and conventional system, the diagnostic methods (RT-PCR) were same. By this study, we just wanted to show the difference of efficiency in time between DT and conventional system, not detection accuracy.

From Feb.26 to Mar.10, how patients were divided into DT or conventional sample collection? Was it random?

Answer: Yes. It`s random.

What is the positive rate for DT group and conventional group during this period.

The evaluation of both efficiency and accuracy are needed.

Answer: We appreciate your valuable comment. Following your comment, we have added positive rate of COVID-19 tests in both DT group and conventional group. However, in both of DT and conventional method, the diagnostic methods (RT-PCR) were totally same. In this study, we just wanted to show the difference of efficiency between DT and conventional method.

1, The evaluation of efficiency includes the cost performance of each procedure in addition to the number of tests. Describe the usage of PPE (personal protective equipment: mask, face shield, gloves, and gown), and compare their per test between DT and conventional specimen collection system.

Answer: We appreciate your valuable comment. In both DT and conventional system, medical staffs wear personal protective equipment (PPE), including N95 respirator, eye–shield/face shield/goggles, hooded coverall/gown, inner and outer gloves continuously, but outer gloves and disposable plastic apron (AP) gowns were changed at every patient. We have added it in the manuscript.

2, The accuracy of DT system may become clear by comparing the positive ratio among the following 4 groups in the 14-day period (from February 26, 2020, to March 10, 2020) during which both systems were operating simultaneously.

Symptomatic-DT

Symptomatic-conventional

Asymptomatic-DT

Asymptomatic-conventional

The DT procedure is accurate when 1)=2), and 3)=4).

Answer: We appreciate your valuable comment. We totally agree with your comment. Following your comment, we have added in detection accuracy of both DT and conventional system in Table 3. However, in both of DT and conventional system, the diagnostic methods (RT-PCR) were same. By this study, we just wanted to show the difference of efficiency in time between DT and conventional system, not detection accuracy.

Reviewer 2 Report

It is an important article related the COVID-19 screening activities. Some recommendations are provided as below:

1. In [Abstract] & [Discussion] section, a powerful conclusion was that the DT system was approximately nine times more efficient at collecting specimens for testing [Line 152-153, Line 28-29]. When using the DT system, the total time spent on each testee was 5-7 minutes [found in Line 103]. However, when using the conventional system, the approximate time (including sterilized time 30 minutes) spent on each testee was not shown (among Line 72-82). How to conclude “the nine times more efficient“ ?

2. Line 129-131: Some mistakes should be collected as below:

(A) Graph showing the “cumulative” number of positive and negative cases …..

(B) Graph showing the “daily” number of positive and negative cases …..

Author Response

Reviewer2

It is an important article related the COVID-19 screening activities. Some recommendations are provided as below:

  1. In [Abstract] & [Discussion] section, a powerful conclusion was that the DT system was approximately nine times more efficient at collecting specimens for testing [Line 152-153, Line 28-29]. When using the DT system, the total time spent on each testee was 5-7 minutes [found in Line 103]. However, when using the conventional system, the approximate time (including sterilized time 30 minutes) spent on each testee was not shown (among Line 72-82). How to conclude “the nine times more efficient“ ?

Answer: We appreciate your valuable comments. We just compared total number of test for COVID-19. Following your comment, we have added more explanation in the manuscript as follows:

In the DT system and the conventional system, the number of specimens collected per day was 241.7 and 41.4 respectively [total numbers: 3,384 (DT system) vs. 580 (conventional system)]. According to total number of test during 14-day period when DT and conventional system were administered simultaneously, the efficiency of the DT system was 5.8 times higher than that of the conventional screening system. In addition, per one test, DT system was 4~6 times faster than conventional system (5-7 minutes vs. 30 minutes).

  1. Line 129-131: Some mistakes should be collected as below:

(A) Graph showing the “cumulative” number of positive and negative cases …..

Answer: We appreciate your valuable comments. Following your comment, we have modified it

(B) Graph showing the “daily” number of positive and negative cases …..

Answer: We appreciate your valuable comments. Following your comment, we have modified it.

Reviewer 3 Report

Thank you for the opportunity to review this paper.  The findings are socially important and timely.  Overall, it is well done.  I have a few suggestions for improving it.  First, sections 3.1.1 and 3.1.2 need more parallel information.  In order to assess the difference between the two beyond total number of tests, it would be helpful to see information in both that allows for comparison.  Just as one example, where the medical professionals in the conventional system wearing PPE?  Similarly, how is payment handled for the DT system?  Finally, the last three sentences in 3.1.2 (lines 101-104) are results/conclusions and seem out of place in the explanations of operational differences.

Author Response

Reviewer 3

Thank you for the opportunity to review this paper.  The findings are socially important and timely.  Overall, it is well done.  I have a few suggestions for improving it.  First, sections 3.1.1 and 3.1.2 need more parallel information.  In order to assess the difference between the two beyond total number of tests, it would be helpful to see information in both that allows for comparison.  Just as one example, where the medical professionals in the conventional system wearing PPE?  Similarly, how is payment handled for the DT system?  Finally, the last three sentences in 3.1.2 (lines 101-104) are results/conclusions and seem out of place in the explanations of operational differences.

Answer: We appreciate your valuable comment. In both conventional system and DT system, all medical staffs wear personal protective equipment, including N95 respirator, eye–shield/face shield/goggles, hooded coverall/gown, inner and outer gloves continuously, but outer gloves and disposable plastic apron (AP) gowns were changed at every patient. We have added it in the manuscript. The operation method of DT and conventional system is different, but the COVID-19 diagnostic method (RT-PCR) and its price are the same.

 We also modified sentence in 3.1.2 (line 101-104) following your comment.

 “Therefore, unlike conventional systems, the DT system does not require spaces to be sterilized between testees and preparation times are markedly reduced. The total time spent on each testee was 5-7 minutes. The DT system is more efficient in time and safer than the conventional screening system.”